# Long-Term Perspective Changes in Crop Irrigation Requirement Caused by Climate and Agriculture Land Use Changes in Rechna Doab, Pakistan

**Arfan Arshad** [1,2,†] **, Zhijie Zhang** [3,†] **, Wanchang Zhang** [1,*] **and Ishfaq Gujree** [2,4]

1 Key Laboratory of Digital Earth Science, Institute of Remote Sensing and Digital Earth, Chinese Academy of Sciences, Beijing 100094, China
2 University of Chinese Academy of Sciences, Beijing 100049, China
3 Department of Geography, Center for Environmental Sciences and Engineering (CESE), University of Connecticut, Storrs, CT 06269-4148, USA
4 Institute of Tibetan Pleatue Research, Chinese Academy of Sciences, Beijing 100094, China
* Correspondence: zhangwc@radi.ac.cn; Tel.: +86-10-82178131
† The first two authors contributed equally to this work and should be considered as co-first authors of this paper.

**Abstract:** Climate change and agriculture land use changes in the form of cropping patterns are closely linked with crop water use. In this study the SDSM (statistical downscaling model) was used to downscale and simulate changes in meteorological parameters from 1961 to 2099 using HadCM3 General Circulation Model (GCM) data under two selected scenarios i.e., H3A2 and H3B2. Results indicated that $T_{max}$, $T_{min}$, and wind speed may increase while relative humidity and precipitation may decrease in the future under both H3A2 and H3B2 scenarios. Downscaled meteorological parameters were used as input in the CROPWAT model to simulate crop irrigation requirement (CIR) in the baseline (1961–1990) and the future (2020s, 2050s and 2080s). Data related to agriculture crop sown area of five major crops were collected from Punjab statistical reports for the period of 1981–2015 and forecasted using linear exponential smoothing based on the historical rate. Results indicated that the cropping patterns in the study area will vary with time and proportion of area of which sugarcane, wheat, and rice, may exhibit increasing trend, while decreasing trend with respect to the baseline scenario was found in maize and cotton. Crop sown area is then multiplied with CIR of individual crops derived from CROPWAT to simulate Net-CIR ($m^3$) in three sub-scenarios S1, S2, and S3. Under the H3A2 scenario, total CIR in S1, S2, and S3 may increase by 3.26 BCM, 12.13 BCM, and 17.20 BCM in the 2080s compared to the baseline, while under the H3B2 scenario, Net-CIR in S1, S2, and S3 may increase by 2.98 BCM, 12.04 BCM, and 16.62 BCM in the 2080s with respect to the baseline. It was observed that under the S2 sub-scenario (with changing agriculture land-use), total CIR may increase by 12.13 BCM (H3A2) and 12.04 BCM (H3B2) in the 2080s with respect to the baseline (1961–1990) which is greater as compared to S1 (with changing climate). This study might be valuable in describing the negative effects of climate and agriculture land use changes on annual crop water supply in Rechna Doab.

**Keywords:** climate change; GCM; SDSM; crop irrigation requirement; effective precipitation; reference ET; CROPWAT

## 1. Introduction

Availability of water resources has great significance for agricultural production, human settlements, industry, and natural vegetation. Nowadays available water resources are being spoiled

as a result of climate as well as land use changes [1]. The continuous change in climatic variables and agricultural expansion has retained new demands on available water resources every day due to changes in crop irrigation requirement [2]. Irrigated agriculture is always ranked 1st in the world according to water consumers as it consumes approximately 64% of fresh water [3–5]. Water withdrawals for agricultural production systems is much focused in space and with respect to crop types. Asian countries especially Pakistan, China, and India, and the United States account for 68% of fresh water withdrawals for irrigated agriculture, out of which ~34% is consumed by India for growing rice and wheat crops covering 70% of the irrigated command area [6].

Many past studies have been conducted for understanding the implications of climate change on crop irrigation requirement (CIR) in various climatic and geographical zones of the world [7–15]. Crop irrigation requirement (CIR) has been largely affected due to changes in the trend of climatic parameters i.e., precipitation, solar radiations, relative humidity, temperature, and wind speed, as a result of climate change [5,16–19]. Studies by different researchers concluded that crop water requirement may increase in most regions of the globe due to increases in Evapotranspiration (ET) and decreases in soil moisture which have happened with warming of the climate [20–22]. Agriculture expansion inversely relates to water consumption of the irrigated command area. In recent years, continuous increment of the irrigated command area has resulted in reduction of the water supply, which will further affect the over abstraction of water resources [23]. Projected results indicated that the future irrigated agriculture area may continuously increase to fulfill food requirements of a rapidly growing population, which will ultimately consume more water regimes to satisfy growing water needs globally [24,25].

Both these issues, expanding agriculture land and changing climate, are more evident in Rechna Doab, Pakistan as it has experienced more variable weather and cropping patterns than other regions in Pakistan. Catchment areas in Indus Basin, Pakistan are accounted as the world's most vulnerable regions to climate change. The food requirement in Pakistan has largely increased as a result of rapid population growth, from 37.5 million in 1950 to 207 million in 2017, and is expected to reach 333 million in 2050 [26]. Land and water resources in Pakistan are tremendously under pressure to fulfill the growing needs of the population [27]. In Rechna Doab, the total irrigated area has increased from 1.94 Mha in 1961 to 2.12 Mha in 1990 and cropping intensity has increased from 91% in 1961 to 131% in 1980 [28]. Indus Basin of Pakistan is allocating irrigated water among four provinces, according to the Indus Water Accord in 1991, and is considered a supply-based system rather than demand-based system. Irrigation water supply is directly affected due to changes in the trend of climatic parameters and the irrigated command area. With the expansion of the irrigated command area in Rechna Doab, irrigation water supply is reducing and end users are not satisfied with available water in the irrigated network [29,30]. Farmers always complain to water authorities (Water and Power Development Authority (WAPDA), Punjab Irrigation and Drainage Authority (PIDA) etc.) about the development of an unequal water distribution system. Therefore, it's important to analyze the impacts of climate and agriculture land use changes on crop irrigation requirement to understand the current and projected changes in crop irrigation requirement (CIR) in Rechna Doab, Pakistan.

Numerous modeling approaches have been used to study the impacts of climate on crop evapotranspiration (ET). Examples: Reference [31] generated future climate using the LARS-WG (Long Ashton Research Station-Weather Generator) in Suwon, South Korea, reference [32] used the generalized linear model (GLM) to simulate future changes in ET, reference [33] applied the Statistical Downscale Model (SDSM) to downscale ET between 2011–2099 from HadCM3 (Hadley Centre Coupled Model, version 3) climate data, reference [34] analyzed the climate change impacts on ET from 1956–2016 using the Penman–Monteith method. The above studies mainly focused on analyzing the spatio-temporal changes in ET caused by the impacts of climate but these studies pay less attention to "how the change of climate would affect the crop water use". Anyhow, some studies analyze perspective changes in crop water requirement due to the changing climate [18,35] and agriculture expansion [36,37]. In these studies, authors analyzed the changes in crop water requirement due to agriculture expansion but they only considered agriculture expansion as one common land-use class i.e., crop land or vegetation.

They did not aggregate agriculture class into various crop-type classes (wheat, sugarcane, rice etc.) and their individual impacts on water requirement based on their proportion of crop sown area in current and future periods. For example, reference [36] analyzed the perspective changes in irrigation requirement caused by agriculture expansion but they did not examine the individual impacts of different agriculture crops on net crop irrigation requirement. Assessing the long-term changes in agriculture land use changes in the form of crop patterns (crop sown area), therefore, became an urgent need to improve decision making processes in water resources management. So, this study analyzed the long-term impacts of climate and agriculture land use (considering the changes in agriculture cropping patter) on crop irrigation requirement in current and future periods.

General circulation models (GCMs) can simulate future changes in the climate on large scale climate studies but they have limitations to their application on small scale studies due to their low spatial resolution. GCM data can be used on the regional scale and small scale studies after downscaling it. CROPWAT developed by FAO (Food and Agriculture Organization) is a better tool to simulate crop irrigation demand in the changing climate [25]. CROPWAT has generally been applied in numerous research studies to compute crop irrigation requirement (CIR) and irrigation scheduling in different countries i.e., Taiwan, Greece, United State, Zimbabwe, Africa, Morocco, Turkey, and Pakistan [38–48]. Aims and objectives of this study were to: (1) To detect long-term trends and changes in climatic parameters (maximum temperature ($T_{max}$), minimum temperature ($T_{min}$), precipitation, humidity, and wind speed); (2) to analyze the long-term changes in agriculture cropping patterns based on crop sown area; and (3) to investigate the implications of climate and agriculture land use variability on total crop irrigation requirement. CROPWAT and the statistical downscale simulation model (SDSM) were coupled with each other to simulate CIR (mm) associated with climate change. Data related to the crop sown area (CSA) of five major crops (wheat, rice, sugarcane, maize, and cotton) was collected from Punjab Statistics Development (PSD) reports for the period of 1981–2015 and forecasted in future periods. Future predicted data of agriculture cropping patterns were integrated with CIR (mm) of individual crops derived from CROPWAT to simulate total CIR ($m^3$) in three different sub scenarios: S1 (climate change), S2 (agriculture land use changes), and S3 (both climate and agriculture land use changes).

## 2. Material and Methods

### 2.1. Study Area

The land exists between two rivers is defined as Doab. Rechna Doab is the part of land situated between two rivers i.e., the rivers Ravi and Chenab in Punjab. It is the most fertile region of the irrigated Indus plain of Punjab and lies in the agro-ecological zone IVa. It starts from Mirpur and Jammu and converges with the total command area (TCA) of 3.0 Mha in which 2.43 Mha area is counted in the irrigated land. It covers eight districts of Punjab, namely Hafizabad, Jhang, Faisalabad, T.T Singh, Faisalabad, Sheikhpura, Gujranwala, Sialkot, and Narowal (Figure 1). It is designated as subtropical semi-arid land and climate of this area shows seasonal fluctuations in precipitation and temperature. The summer season is considered as long and hot which starts from April and ends in September with Tmax 33–48 °C. The winter season starts from December and ends in February, with Tmax 19–27 °C. Annual precipitation in the study area is about 655 mm in the upper side of Rechna, and 360 mm in lower part of Rechna Doab. Surface canal irrigation system of this doab is gravity flow and perennial canals flow at a normal period, about 340 days per year.

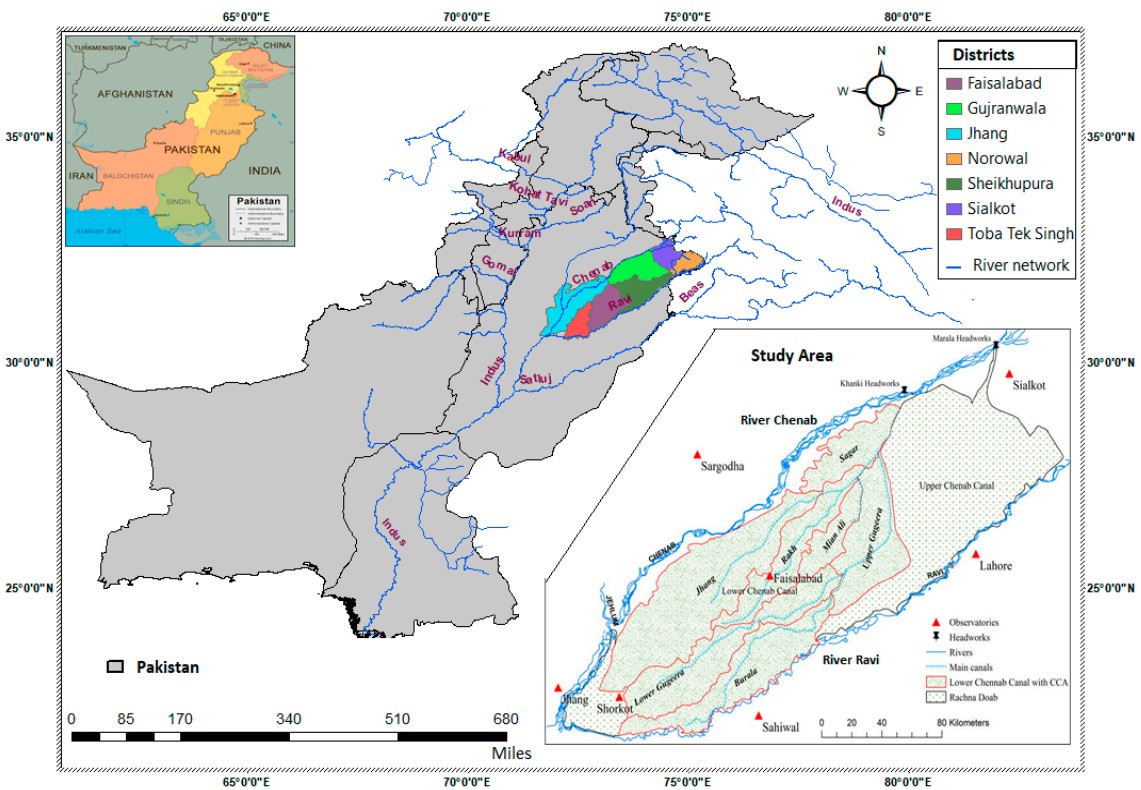

**Figure 1.** Map of Rechna Doab, Pakistan, configuration of irrigation network, irrigation districts, head-works, and metrological observatories located in the study area.

*2.2. Data Used in Study*

Data used in this study comprises of meteorological data (temperature, humidity, wind speed, precipitation, sunshine hours, and sun radiations), GCMs data (National Center of Environmental Prediction–National Center for Atmospheric Research (NCEP) (Table 1), and Hadley Centre coupled model (HadCM3)) and crop information data.

**Table 1.** List of 26 NCEP (National Centers for Environmental Prediction) predictors used in this study.

| No | Predictors | Code | No | Predictors | Code |
|----|-----------|------|----|-----------|------|
| 1 | Mean sea level pressure | Mslpas | 14 | 850 hPa air flow strength | P5zhas |
| 2 | Surface flow strength | P_fas | 15 | 850 hPa zonal velocity | P8_fas |
| 3 | Surface zonal velocity | P_uas | 16 | 850 hPa meridian velocity | P8_uas |
| 4 | Surface meridional velocity | P_vas | 17 | 850 hPa vorticity | P8_vas |
| 5 | Surface vorticity | P_zas | 18 | 850 hPa wing direction | P8_zas |
| 6 | Surface wind direction | P_thas | 19 | 850 hPa divergence | P850as |
| 7 | Surface divergence | P_zhas | 20 | 500 hPa geopotential height | P8thas |
| 8 | 500 hPa Air flow strength | P5_fas | 21 | 850 hPa geopotential height | P8zhas |
| 9 | 500 hPa zonal velocity | P5_uas | 22 | Near surface relative humidity | R500as |
| 10 | 500 hPa meridional velocity | P5_vas | 23 | Near surface specific humidity | R850as |
| 11 | 500 hPa vorticity | P5_zas | 24 | 500 hPa specific/relative humidity | Rhumas |
| 12 | 500 hPa wind direction | P500as | 25 | 850 hPa specific/relative humidity | Shumas |
| 13 | 500 hPa divergence | P5thas | 26 | Mean temperature at 2 m | Tempas |

2.2.1. Meteorological Data

The Punjab Meteorological Department provided daily meteorological data including maximum temperature, minimum temperature (Tmax, Tmin, °C), relative humidity (RH, %), wind speed (WS, m s$^{-1}$), sunshine hours (SH, h), and precipitation. Observed climate data was collected for five

meteorological stations i.e., Faisalabad, Lahore, Sargodha, Jhelum, and Sialkot meteorological from 1980–2014 at daily scale. These datasets were used to calibrate and validate the statistical downscale simulation model (SDSM).

### 2.2.2. GCM Data

The GCM used in this study was derived from HadCM3, the latest version model with improved ocean and atmosphere components [49]. It comprises of daily predictor variables of NCEP 1961–2000, H3A2a 1961–2099, and H3B2a 1961–2099. H3A2 and H3B2 data was used to simulate changes in future scenarios. NCEP data was utilized for calibration and validation of the SDSM model against the observed data. HadCM3 is a coupled climate model with spatial resolution of 2.50 latitude and 3.750 longitude, and freely downloadable from http://www.cics.uvic.ca/scenarios/sdsm/select.cgi. HadCM3 data under two IPCC emission scenarios i.e., H3A2 and H3B2 was collected for the period of 1961 to 2099. These emission scenarios have already developed by the IPCC (Inter-Governmental Panel on Climate Change) [50].

### 2.2.3. Crop Information Data

For computation of CIR, CROPWAT requires multi crop parameters i.e., planting and harvesting dates of crops, crop coefficient (Kc), rooting depth, allowable depletion, and yield response. Information of these parameters was collected from relevant studies, i.e., cropping pattern information collected from paper of reference [51], planting and harvesting data was collected from past research [52,53] and Kc values, crop rooting depth, allowable depletion, and yield response factor were taken from FAO (Table 2). Table 2 illustrates the main information of five major crops i.e., sugarcane, maize, wheat, rice, and cotton, growing in the study area. Statistics data of the crop sown area of five major crops grown in eight districts (Faisalabad, Hafizabad, Jhang, T.T Singh, Sheikhu-Pura, Gujranwala, Sialkot, and Narowal) were collected for the period of 1981–2008 from the Government of Pakistan statistics reports [54] and 2009–2015 from Punjab Development statistics (PDS) reports (Figure 2).

**Table 2.** Multi crop information parameters used in the CROPWAT model to compute Crop Irrigation Requirement (CIR) of five major crops grown in Rechna Doab.

| Crop Types | Date (Sowing-Harvesting) | Stages | Period | $K_c$ | Rooting Depth | Crop Height | Yield Response Factor | Depletion Factor |
|---|---|---|---|---|---|---|---|---|
| | | | days | | m | m | | |
| Sugarcane | Annual Jan-Dec | Initial | 30 | 0.40 | 1.5 | | 0.50 | 0.65 |
| | | Development | 60 | | | 3 | 0.75 | |
| | | Mid-season | 180 | 1.25 | | | 1.20 | 0.65 |
| | | Late season | 95 | 0.75 | 1.5 | | 0.10 | 0.65 |
| Wheat | Nov-Mar | Initial | 20 | 0.4 | 0.30 | | 0.20 | 0.55 |
| | | Development | 30 | | | 0.8 | 0.69 | |
| | | Mid-season | 50 | 1.15 | | | 0.50 | 0.55 |
| | | Late season | 30 | 0.41 | 1.50 | | 0.40 | 0.90 |
| Rice | June-Nov | Initial | 20 | 1.05 | 0.10 | | 1.00 | 0.20 |
| | | Development | 30 | | | 1 | 1.09 | |
| | | Mid-season | 40 | 1.20 | | | 1.31 | 0.20 |
| | | Late season | 30 | 0.99 | 0.60 | | 0.50 | 0.20 |
| Cotton | May-Nov | Initial | 30 | 0.35 | 0.30 | | 0.20 | 0.65 |
| | | Development | 50 | | | 1.30 | 0.59 | |
| | | Mid-season | 60 | 1.15 | | | 0.50 | 0.65 |
| | | Late season | 55 | 0.5 | 1.40 | | 0.25 | 0.90 |
| Maize | May-Sep | Initial | 20 | 0.30 | 0.30 | | 0.40 | 0.55 |
| | | Development | 35 | | | 1.5 | 0.40 | |
| | | Mid-season | 40 | 1.2 | | | 1.30 | 0.55 |
| | | Late season | 30 | 0.35 | 1 | | 0.50 | 0.80 |

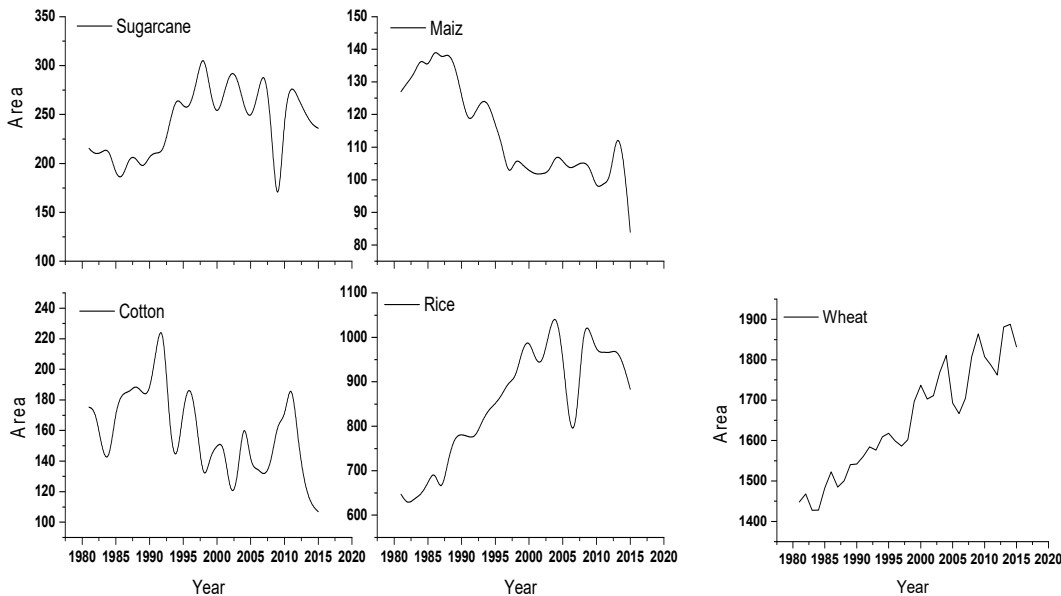

**Figure 2.** Long-term changes in crop sown area (CSA) (1000 hectare) of five major agricultural crops grown in Rechna Doab from 1981–2015.

### 2.3. Downscaling of GCM Data Using SDSM Model

Downscaling is a technique of improving resolution of GCMs data for its use at the local scale. Downscaling approach can be adapted on spatio-temporal features of the climate system [55–57]. Dynamical and statistical are two downscaling approaches which are widely used for downscaling of GCM data [55,58–60]. The dynamical downscaling technique uses complex algorithms to generate fine spatial resolution climatic information supplied by GCMs at coarse resolution approximately 20–50 km. Dynamical downscaling is a process of nesting fine spatial resolution data of the regional climate model (RCMs) within low resolution data of GCMs [61]. The statistical downscaling approach generates future climate scenarios by developing statistical relationships between global scale features and local variables [62–64]. The statistical downscaling approach is preferred rather than the dynamical downscaling approach because of its simple computation and is easily adapted to new spaces [55,65–67]. Therefore, the statistical downscaling approach was used in this study to downscale GCMs data under two scenarios, A2 and B2.

SDSM was developed by Wilby, Dawson and Barrow [67], and it uses multi linear regression (MLR) to establish statistical relationships between large scale NCEP predictors and local-scale predictands [68]. SDSM provides two types of sub-models, (1) unconditional model for local variables distributed normally such as temperature, and (2) conditional sub model is applied when local variables are not distributed normally such as precipitation and evaporation [69]. SDSM can normalize such variables by transforming parameters using different functions i.e., natural log, fourth root, and inverse normal [64,70,71]. In this study, the unconditional sub-model with no transformation function was applied to downscale T_max, T_min, wind speed, and relative humidity while in the case of precipitation conditional sub-model, was applied with transformation function (fourth root). In order to downscale GCMs data, the following steps were adapted: (1) Quality control analysis for verification of missing data values, (2) screening of variables using NCEP predictor, (3) calibration of the model, (4) weather generation, (5) validation of SDSM, and (6) generation of future climate scenarios using GCMs.

(1) *Screening of NCEP Predictors:* Screening of NCEP predictor variables is a crucial step for all statistical downscaling techniques because these parameters greatly influences the output of the model [71,72]. In this study, during screening process correlation between 26 NCEP predictors (Table 1) and local scale predictands (observed precipitation, temperatures, wind speed, and relative humidity) was developed in SDSM model, and then the predictors of highest correlation

coefficient among 26 predictors were finally selected. (Table 3). Predictors with low $R^2$ and highest *P* value (greater than α (0.05)) were neglected to minimize uncertainty in future prediction. The highest value of $R^2$ as 0.7 is satisfactory in calibration and validation of the SDSM model [73].

(2) *Calibration and Validation of model:* NCEP predictor variables having highest value of $R^2$ were used in the weather generation process. Observed data of climatic parameters was divided into two halves, the first part (1980–1989) was used to calibrate the SDSM model and the second part (1990–1999) was used to validate the model. During the calibration period (1980–1989) and validation period (1990–1999), simulated results of SDSM were compared with the observed data of Tmax, Tmin, humidity, wind speed, and precipitation.

(3) *Scenario Generation:* After successful calibration and validation of SDSM, future scenarios were generated using HadCM3 data under A2 and B2 scenarios within the time span of 1961 to 2099. Three time windows, 2020 (2010–2039), 2050 (2040–2069), and 2080 (2070–2099) were constructed to assess the patterns of climate variables in different spans.

**Table 3.** Screening of most appropriate NCEP predictor variables depicted good correlation with observed climate parameters.

| No. | Predictors | Code | $T_{max}$ | $T_{min}$ | Precp | R.H | WDS |
|-----|------------|------|-----------|-----------|-------|-----|-----|
| 1 | Mean sea level pressure | Mslpas | √ | √ | √ | √ | √ |
| 3 | Surface zonal velocity | P_uas | √ | √ | √ | | √ |
| 5 | Surface vorticity | P_zas | √ | | √ | √ | |
| 8 | 500 hPa Air flow strength | P5_fas | √ | | | √ | |
| 11 | 500 hPa vorticity | P5_zas | √ | | | | √ |
| 12 | 500 hPa wind direction | P500as | √ | | | √ | √ |
| 14 | 850 hPa air flow strength | P5zhas | √ | √ | √ | | √ |
| 16 | 850 hPa meridian velocity | P8_uas | √ | √ | √ | √ | |
| 17 | 850 hPa vorticity | P8_vas | √ | √ | | | |
| 18 | 850 hPa wing direction | P8_zas | √ | √ | √ | | √ |
| 19 | 850 hPa divergence | P850as | √ | | | √ | √ |
| 20 | 500 hPa geopotential height | P8thas | | | √ | √ | |
| 21 | 850 hPa geopotential height | P8zhas | √ | | √ | √ | |
| 23 | Near surface specific humidity | R850as | | | √ | √ | |
| 25 | 850 hPa specific/ relative humidity | Shumas | | √ | √ | √ | √ |
| 26 | Mean temperature at 2 m | Tempas | √ | √ | √ | √ | √ |

### 2.4. Prediction of Agriculture Cropping Patterns

Data related to crop sown area were collected from Punjab Statistical Report (PSR) and forecasted in the future using linear and exponential forecasting methods, considering highest values of $R^2$ based on historical rate. These methods can be used to predict future changes [74]. Agriculture land use changes in the form of cropping patterns (sown area) of different agriculture crops were observed in the baseline (1980–2015) and future time windows (2020s, 2050s, and 2080s). This data was linked with CROPWAT to check their impacts on total irrigation requirement for each crop.

### 2.5. Modeling Crop Irrigation Requirement (CIR)

CIR (crop irrigation requirement) is the quantity of water required to fulfil the evapotranspiration loss and leaching of salts from a cropped field. In this study CIRs of five major agricultural crops (wheat, sugarcane, rice, cotton, and maize) were computed using the CROPWAT model developed by FAO (Food Agriculture Organization). Meteorological variables (Tmax, Tmin, precipitation, humidity, sunshine hours, and sun radiations) downscaled from SDSM were used as input in the CROPWAT

model to compute CIR in the baseline (1961–1990) and future scenarios (202s, 2050s, and 2080s). In this study the following relations were used to compute CIR (mm).

$$ET_c = K_c \times ET_o \tag{1}$$

$$CIR = ETc - PE \tag{2}$$

where *CIR* is crop irrigation requirement in mm, *ETc* represents the crop evapotranspiration (mm day$^{-1}$), *PE* the amount of effective precipitation (mm), $ET_o$ is the reference *ET* (mm day$^{-1}$) and $K_c$ is the crop coefficient. Reference *ET* was computed using CROPWAT which applies Penman–Monteith equation recommended by FAO [75]. $ET_o$ can be defined as amount of water transpired by well grown grass with height 0.12 m and albedo of 0.23. The following relation was used to estimate reference *ET*

$$ET_o = \frac{0.408\Delta(R_n - G) + u\frac{900}{T+273}u_2(e_s - e_a)}{\Delta + u(1 + 0.34u_2)} \tag{3}$$

where $R_n$ is the net amount of radiations (MJ m$^{-2}$ day$^{-1}$), *G* is density of heat flux in soil (MJ m$^{-2}$ day$^{-1}$), u$_2$ is the mean wind speed in 24 h (m s$^{-1}$), $e_s$ is the saturation vapor pressure (kPa), $e_a$ is actual vapor pressure (kPa), $\Delta$ is gradient of vapor pressure vs. temperature curve (kPa °C$^{-1}$), and *u* is the psychrometric constant (kPa °C$^{-1}$) [76]. The effective rainfall in agricultural region is defined as the amount of precipitation infiltrated into the soil, stored in root zone, and later it transpired by crops. CROPWAT computed effective precipitation ($P_{eff}$) using the soil conservation service method proposed by the United States Department of Agriculture (USDA).

$$P_{eff} = \frac{(P \times (125 - 0.2 \times 3P))}{125} \quad \text{For } P \leq 250/3 \text{ mm} \tag{4}$$

$$P_{eff} = \frac{125}{3} + 0.1P \quad \text{For } P > 250/3 \text{ mm} \tag{5}$$

where *P* is amount of precipitation (mm).

*2.6. CIR under Changing Climate and Agriculture Cropping Patterns*

CIR of individual crops derived from CROPWAT model and agriculture crop sown area were integrated in Equation (6) to compute total CIR in the baseline (1961–1990) and future scenarios (2020s, 2050s, and 2080s).

$$\text{Total CIR} = \sum_{i=1}^{n} A_i \times CIR_i \tag{6}$$

where $CIR_i$ stands for crop irrigation requirement of crop *i*, $A_i$ is the crop sown area (CSA) of crop *i*. In order to assess individual and integrated impacts of climate and agriculture cropping patterns, future changes in CIR were assessed in three different sub-scenarios (1) S1: Changed climate with no change in agriculture land use, (2) S2: Changed agriculture land use with no change in climate, (3) S3: Changed climate and agriculture land use. Figure 3 depicts the stepwise procedure of data analysis.

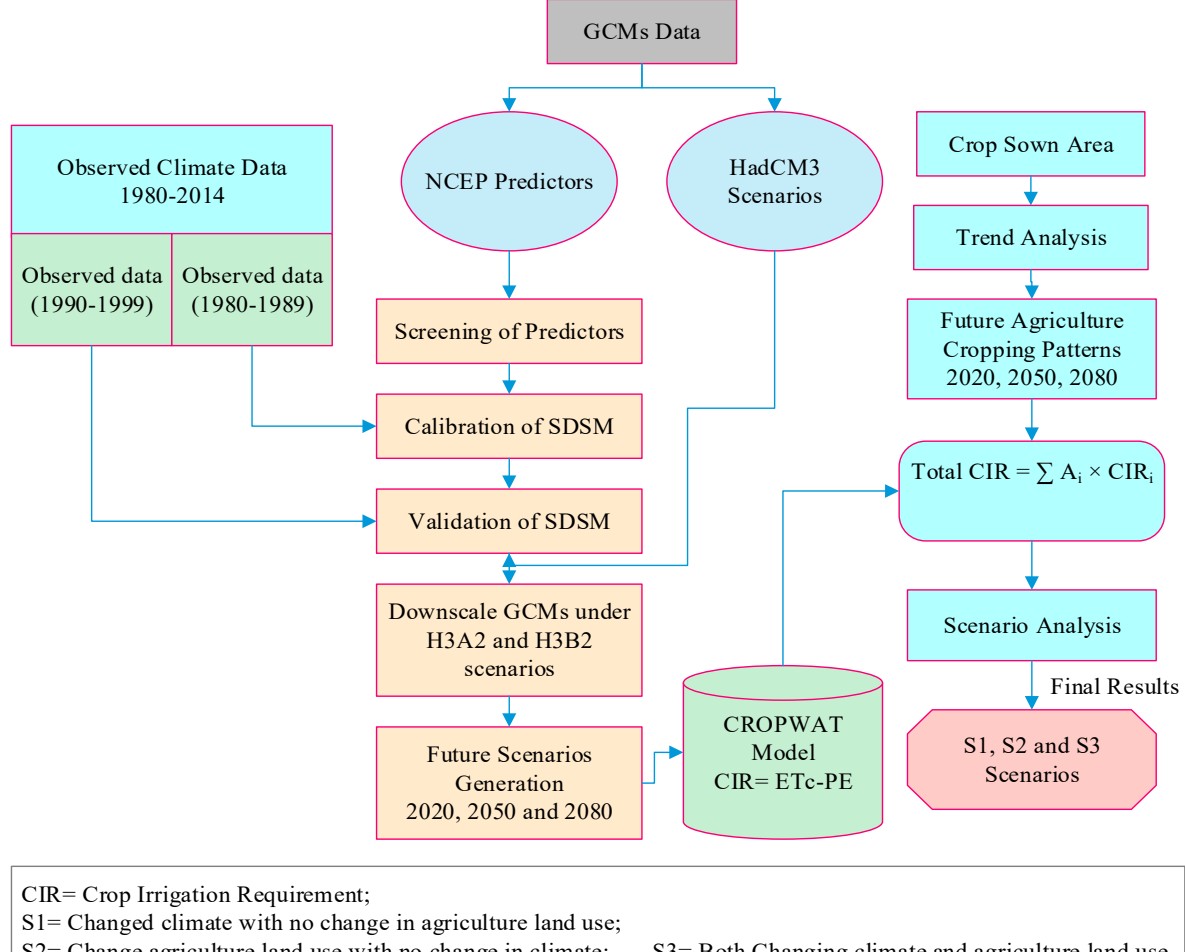

**Figure 3.** Stepwise procedure of computing crop irrigation requirement (CIR) using observe climate data, General Circulation Models (GCMs), and crop information data.

## 3. Results

### 3.1. Screening of NCEP Predictor Variables

Table 3 indicates the most appropriate NCEP predictors screened out having super correlation with the observed local climate predictands (Tmax, Tmin, precipitation, relative humidity, and wind speed). It was observed that NCEP temperature (temperature at 2 m height) has super correlation with Tmax and Tmin. Therefore, this is the super predictor for downscaling Tmax and Tmin. Similar results were predicted in the study conducted by Mahmood and Babel [64], in which they used NCEP temperature (temperature at 2 m height) for downscaling Tmax and Tmin due to its significant correlation with observed metrological parameters. For precipitation, two predictors were found with super correlation i.e., 850 hPa meridian velocity and 500 hPa vorticity. In the case of relative humidity, different predictors were selected, however the most correlated super predictor was found as 500 hPa wind direction. Mean sea level pressure (pmlspas) was found as the super predictor for wind speed. The predictors selected for local climate parameters are mostly the same as used in previous studies [69,71,77].

### 3.2. Calibration and Validation of SDSM Model

NCEP predictors having super correlation with local climate parameters were finally selected and used in weather and scenario generation. Simulated results obtained from SDSM during the

calibration (1980–1989) and validation (1990–1999) period were compared with observed data. During the calibration and validation period, simulated Tmax, Tmin, wind speed, relative humidity, and precipitation were found highly correlated with the observed data. During the calibration period, values of $R^2$ for Tmax, Tmin, wind speed, relative humidity, and precipitation were 0.99, 0.99, 0.87, 0.98, and 0.98, respectively, while in the case of the validation period these were 0.99, 0.99, 0.76, 0.99, and 0.97, respectively (Figure 4). Higher value of $R^2$ indicated that the SDSM model is reliable and very applicable in Rechna Doab, Pakistan for climate change prediction.

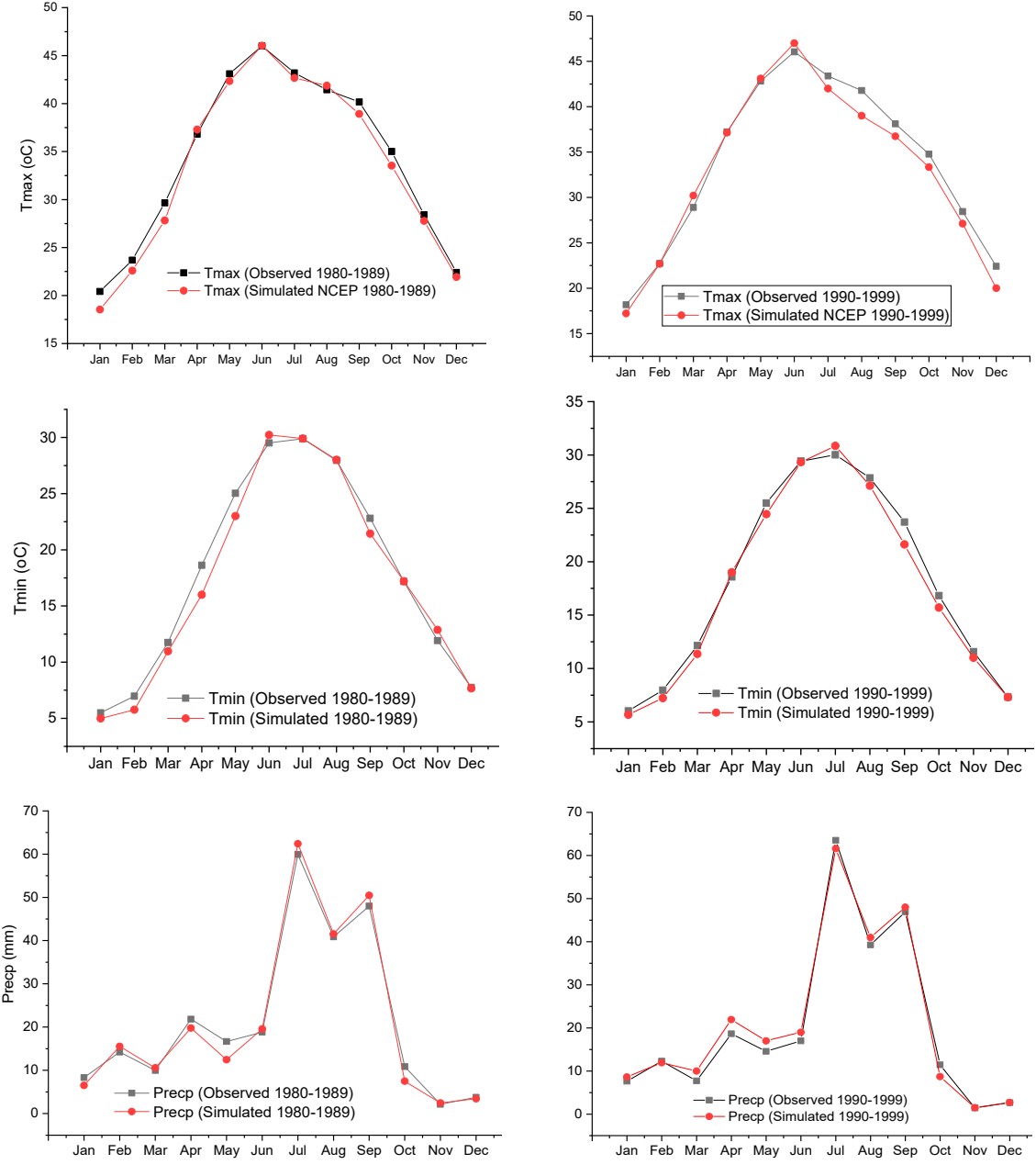

**Figure 4.** *Cont.*

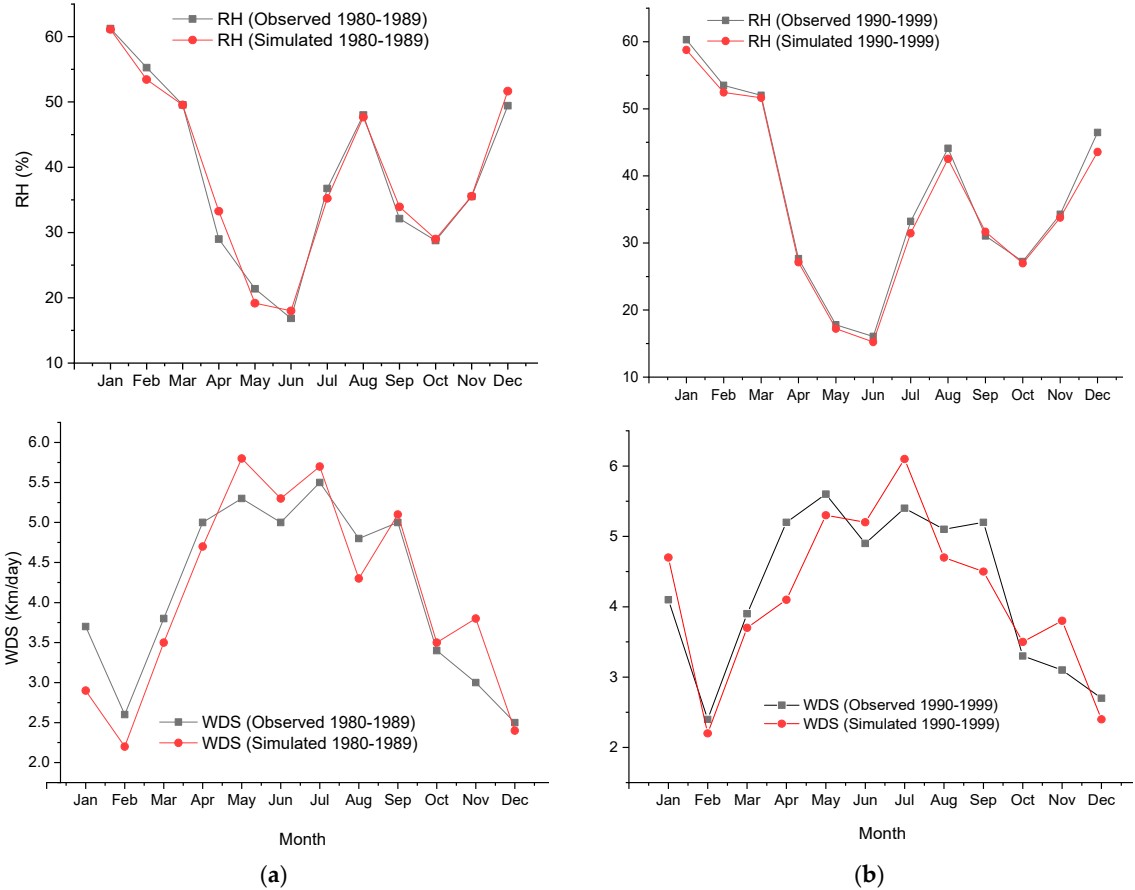

**Figure 4.** Comparison of observed and modeled (NCEP) mean monthly Tmax (maximum temperature, Tmin (minimum temperature), Precp (precipitation), WDS (Wind speed), and R.H (relative humidity) during (**a**) calibration period (1980–1989) and (**b**) validation period (1990–1999).

### 3.3. Predicted Changes in Local Climate of Rechna Doab

This paper investigated the long-term variations in climate change parameters (temperature, precipitation, relative humidity, and wind speed) within the study area in the baseline (1961–1990) and future (2020s, 2050s, and 2080s) under two selected scenarios i.e., H3A2 and H3B2. Figure 5 presented downscaled monthly values of climate change parameters under (a) H3A2 and (b) H3B2 scenarios. Table 4 illustrated the mean values of climatic parameters in the different time period. Long-term changes indicated that in the H3A2 scenario, the average annual Tmax, Tmin, and wind speed may increase from 33.54 to 37.05 °C, 17.13 to 19.74 °C, 97.38 to 115.3 km h$^{-1}$ respectively, while relative humidity and precipitation may decrease from 31.18% to 34.28%, and 262.60 to 170.59 mm in the 2080s with respect to the baseline scenario (1961–1990). In case of the H3B2 scenario, the average annual Tmax, Tmin, and wind speed may increase from 33.39 to 36.28 °C, 16.98 to 19.39 °C, 93.2 to 111.4 km h$^{-1}$ respectively, while relative humidity and precipitation may decrease from 37.24% to 35.51% and 261.95 to 163.97 mm in 2080s with respect to the baseline scenario (1961–1990). It was observed that temperature and wind speed may exhibit an upward trend while precipitation and humidity may exhibit a downward trend in the study area with respect to the baseline period. Analysis indicated that the future local climate of Rechna Doab, Pakistan would be warmer and dryer. The A2 scenario exhibits more of an increase in temperature compared to the B2 scenario. SDSM results indicated that all seasons may exhibit a rise in temperatures (Tmax, Tmin) but the maximum rise is found in the autumn season (September–November) under both scenarios H3A2 and H3B2. Precipitation is expected to decrease in all seasons (except winter) relative to the baseline, but the maximum decrease in precipitation may occur to appear in the spring season with 63 mm in H3A2 and 53 mm in H3B2.

The gradual rise in temperature and reverse trend in precipitation may have significant implications on crop water demand in this area.

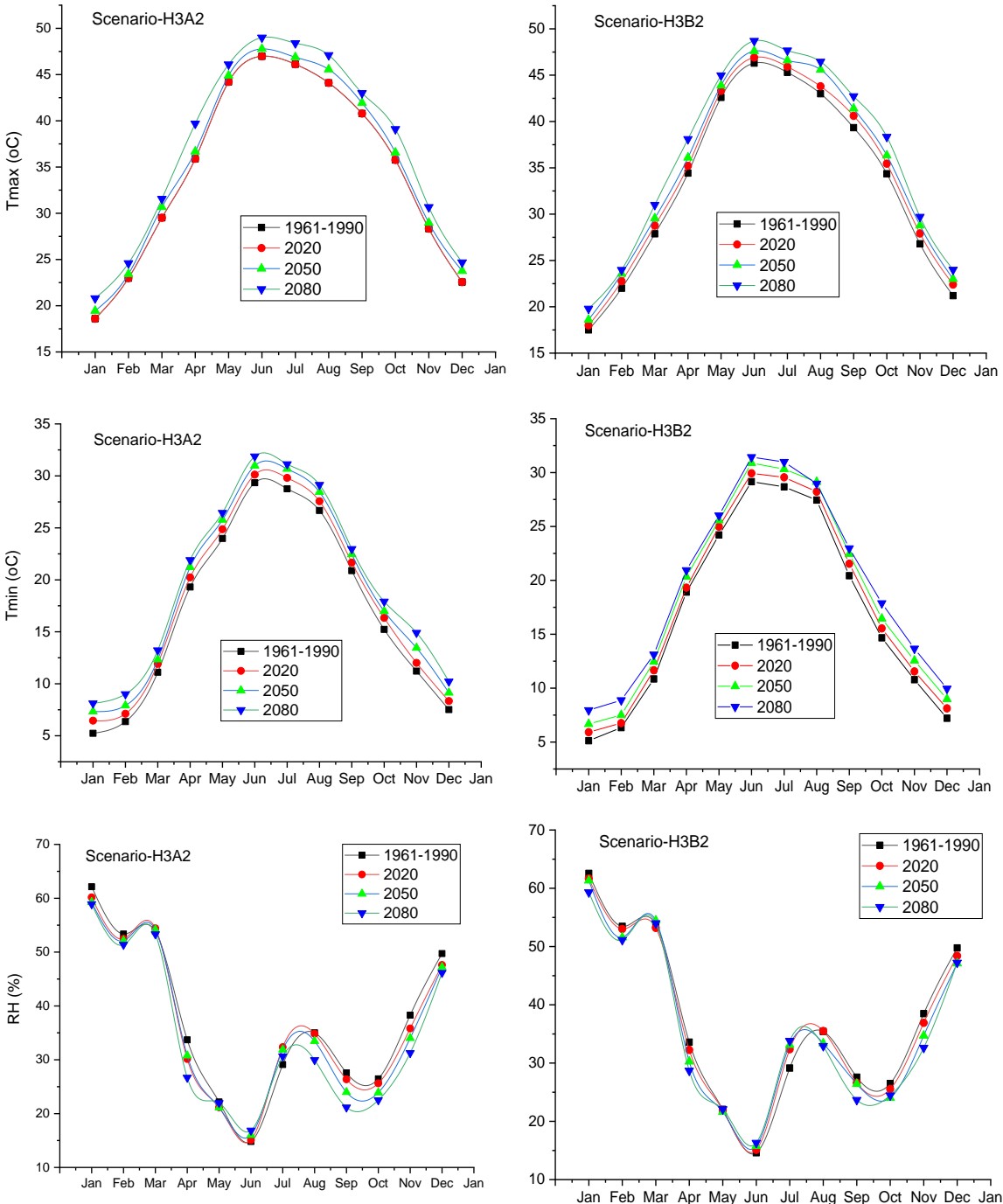

**Figure 5.** *Cont.*

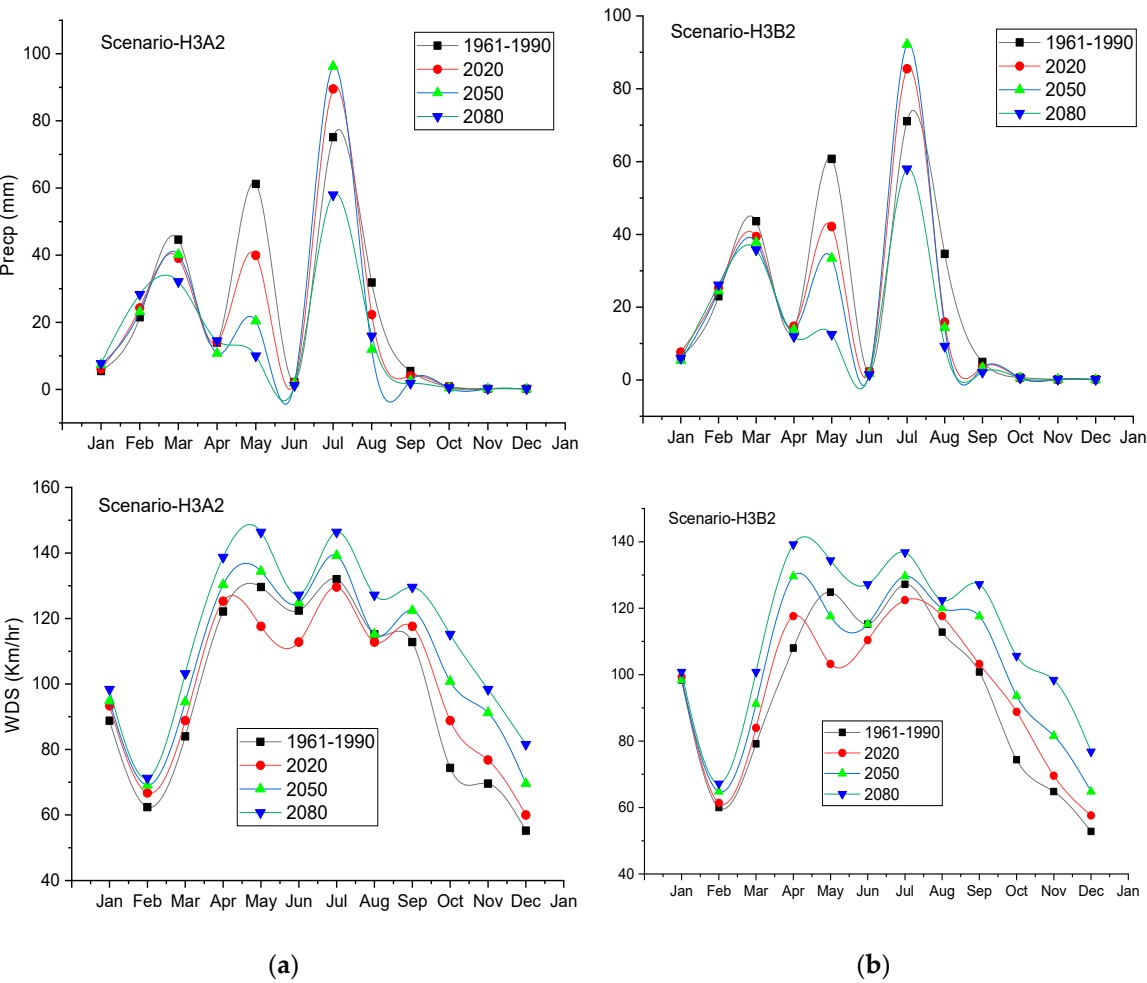

**Figure 5.** Downscaled mean monthly values of climatic parameters (mean monthly Tmax (maximum temperature, Tmin (minimum temperature), Precp (precipitation), WDS (wind speed), and R.H (relative humidity)) in the baseline (1961–1990) and future (2020s, 2050s, and 2080s) under (**a**) H3A2 and (**b**) H3B2, scenarios.

**Table 4.** Relative changes in average values of climatic parameters in future periods with respect to the baseline period (1961–1990).

| Scenarios | Tome Period | Tmax °C | Tmin °C | RH % | Precp mm | Wind Speed km h$^{-1}$ |
|---|---|---|---|---|---|---|
| H3A2 | Baseline (1961–1990) | 35.54 | 17.13 | 31.18 | 262.60 | 97.38 |
| | 2020 | 34.65 | 18.03 | 36.33 | 242.22 | 99.18 |
| | 2050 | 35.54 | 18.89 | 35.65 | 214.85 | 107.2 |
| | 2080 | 37.05 | 19.74 | 34.28 | 170.59 | 115.3 |
| H3B2 | Baseline (1961–1990) | 33.39 | 16.98 | 37.24 | 261.50 | 93.2 |
| | 2020 | 34.32 | 17.75 | 36.87 | 236.95 | 94.59 |
| | 2050 | 35.08 | 18.60 | 36.14 | 223.27 | 102 |
| | 2080 | 36.28 | 19.39 | 35.51 | 163.97 | 111.4 |

## 3.4. Predicted Changes in Reference ET

Future changes in meteorological parameters may have greater implications for reference ET ($ET_o$). Figure 6 describes variations in monthly $ET_o$ in baseline (1961–1990) and future (2020s, 2050s, and 2080s) under two selected scenarios (a) H3A2 and (b) H3B2. Under both selected scenarios, $ET_o$ is increasing in all months from 2020 to 2080, however, maximum increment was found from May to September. Monthly variations of $ET_o$ indicated that, in the H3A2 scenario, it may increase gradually

from January as 1.67 (baseline), 1.78 (2020s), 1.85 (2050s), and 1.95 (2080s) mm day$^{-1}$ to peak value of about 7.29 (baseline), 7.42 (2020s), 7.82 (2050s), 8.01 (2080s) mm day$^{-1}$ in July and then it decreases gradually to 1.64 (baseline), 1.73 (2020s), 1.9 (2050s), and 2.09 (2080s) mm day$^{-1}$ in December. The same variation trend was found in the H3B2 scenario. Under both scenarios, monthly $ET_o$ increased gradually from January to the peak value in June, July and then decreased gradually up to December. Under the H3A2 scenario, average annual $ET_o$ in the baseline period (1961–1990) was 4.55 mm day$^{-1}$ and in the future it may reach up to 4.70 mm day$^{-1}$ in the 2020s, 4.90 mm day$^{-1}$ in the 2050s, and 5.17 mm day$^{-1}$ in the 2080s, while under the H3B2 scenario the average annual $ET_o$ in the baseline was 4.43 mm day$^{-1}$ and in the future it may reach up to 4.57 mm day$^{-1}$ in the 2020s, 4.76 mm day$^{-1}$ in the 2050s, and 5.03 mm day$^{-1}$ in the 2080s. ETO would gradually increase by 3.1% in the 2020s, 7.14% in the 2050s, and 12% in the 2080s in the H3A2 scenario while in the H3B2 it would increase by 3.12% in the 2020s, 6.96% in the 2050s, and 11.92% in the 2080s with respect to the baseline period. The H3A2 scenario may exhibit greater rise in reference ET compared to the H3B2 scenario due to a greater rise in temperature in the H3A2 scenario. Seasonal changes indicate that under both scenarios, autumn may exhibit maximum increase in ET0 in the future periods relative to the baseline, which is due to a greater rise in temperature and low rainfall in this season. Overall, with the changes of climatic factors, $ET_o$ in Rechna Doab may continue to increase in the future, which would cause a gradual rise in CIR.

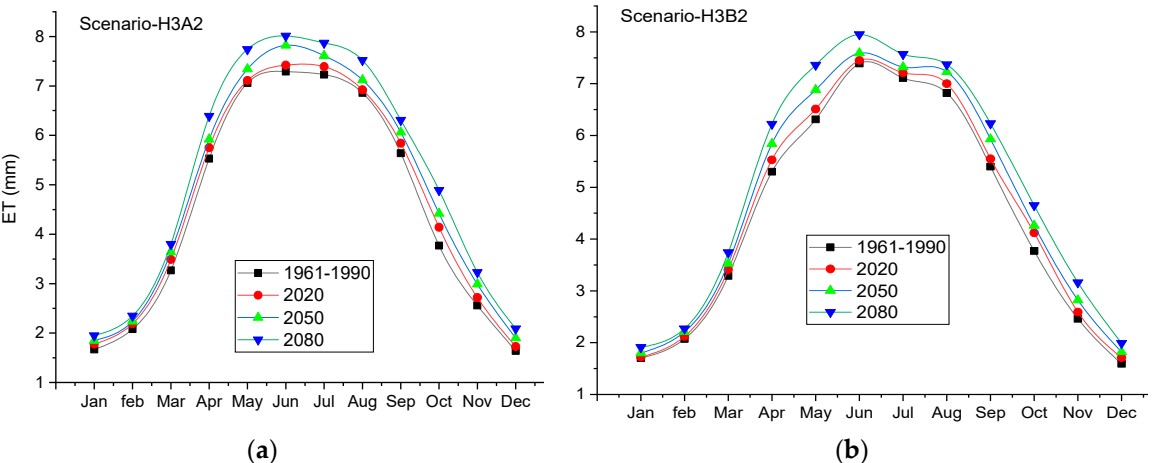

**Figure 6.** Predicted changes in mean monthly evapotranspiration (mm day$^{-1}$) in the baseline (1961–1990) and the future (2020s, 2050s, and 2080s) under (**a**) H3A2 and (**b**) H3B2 scenarios.

### 3.5. Predicted Changes in Effective Precipitation (Pe)

It can be seen from Figure 7 that the monthly effective precipitation (Pe) under both scenarios (a) H3A2 and (b) H3B2 show downward trends in future periods. The effective precipitation showed considerable monthly variation. The maximum effective precipitation was in July, whereas it was close to zero in October–December. Results indicated that average annual Pe may decrease from 210 mm in the baseline (1961–1990) to 193 mm in the 2020s, 172 mm in the 2050s, and 136 mm in the 2080s under H3A2 scenario. While under the H3B2 scenario it may decreased from 209 mm in the baseline (1961–1990) to 189 mm in the 2020s, 181 mm in the 2050s, and 131 mm in the 2080s. Annual Pe may decrease by 7.76% in the 2020s, 18.18% in the 2050s, and 35.03% in the 2080s in the H3A2 scenario while in H3B2 it would decrease by 9.38% in the 2020s, 13.07% in the 2050s, and 37.29% in the 2080s with respect to the baseline period. Seasonal variations indicated that Pe may exhibit a downward trend in the future except winter which may exhibit an increment in Pe which is due to the rise in precipitation in this season. Due to combined impact, an increase in ET and decrease in PE may have greater implications for irrigation requirement and it could rise in the future.

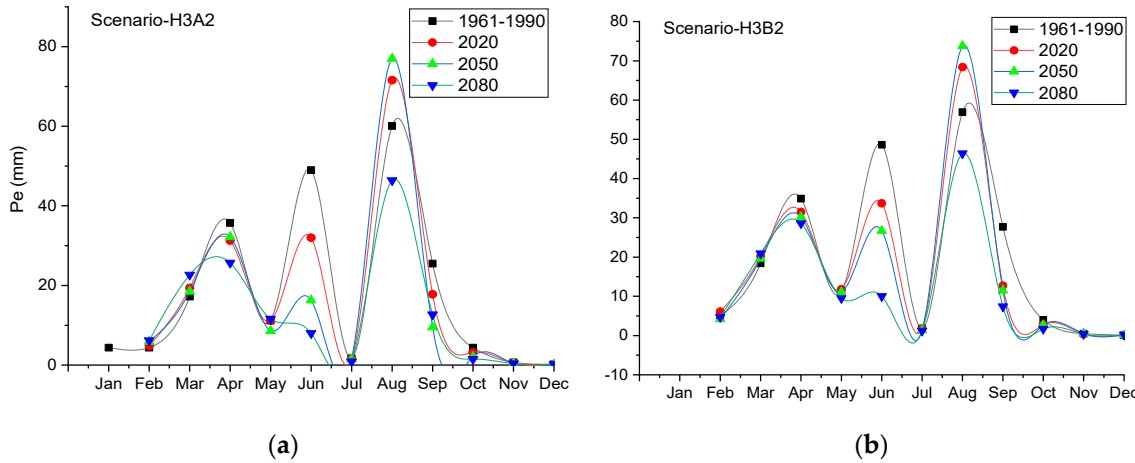

**Figure 7.** Mean monthly changes in effective precipitation (mm) in the baseline (1961–1990) and the future (2020s, 2050s, and 2080s) under (**a**) H3A2 and (**b**) H3B2 scenarios.

*3.6. Predicted Changes in Agriculture Cropping Patterns*

Table 5 depicted changes in agriculture land use changes in the form of cropping patterns (crop sown area) of five major crops, wheat, sugarcane, rice, cotton, and maize grown on a large area of Rechna Doab. Results indicated that in future scenarios, crop sown area of sugarcane, wheat and rice may exhibit an increasing trend while cotton and maize exhibit a reducing trend. In the baseline scenario (1981–2015), total crop sown area of five major agriculture crops was $3011.23 \times 10^3$ ha and in the future it may reach to $3274.5 \times 10^3$ ha (2020s), $3693.5 \times 10^3$ ha (2050s), and $4220 \times 10^3$ ha (2080s). Wheat crop may exhibit a rapid expansion in the cultivated area as it was grown on $1648.68 \times 10^3$ ha in the baseline period (1981–2015) and in the future it may reach up to $1900 \times 10^3$ ha (2020), $2250 \times 10^3$ ha (2050), and $2448 \times 10^3$ ha (2080s). Wheat grown on a large area of Rechna Doab is followed by rice and sugarcane.

**Table 5.** Future changes in agriculture crop sown area of five major crops (unit: 1000 ha).

| Crop Type | Baseline (1981–2015) | 2020 | 2050 | 2080 |
|---|---|---|---|---|
| Sugarcane | 241.55 | 270 | 308.5 | 351 |
| Wheat | 1648.68 | 1900 | 2250 | 2448 |
| Rice | 852 | 890 | 1400 | 1710 |
| Cotton | 156 | 120 | 80 | 60 |
| Maize | 113 | 64 | 66 | 47 |
| Total | 3011.23 | 3274.5 | 3693.5 | 4220 |

*3.7. Predicted Changes in CIR*

ETc is greatly correlated with $ET_o$, and represents the amount of water to irrigate the crops (i.e., CIR in this case). Summer and spring seasons exhibit maximum rate of $ET_O$ which indicates that crops grown in these seasons may consume more water than other seasons. Moreover, some crops may have their development stage during these seasons which may have high value of Kc and result in consumption of more water. Figure 8 depicts changes in the monthly CIR (mm) of five major agricultural crops under two selected scenarios (a) H3A2 and (b) H3B2. CIR of each crop may exhibit an increasing trend in each month from the baseline period (1961–1990) to the future (2020s, 2050s, and 2080) under both selected scenarios i.e., A2 and B2, but with different magnitudes. These changes are cross-ponding to rise in temperature and decrease in precipitation. Figure 9 explains the variations of monthly total CIR of five crops under both scenarios. Under the H3A2 scenario, the monthly total CIR of five agricultural crops was highest in August with 911 mm in the baseline (1961–1990), 939 mm in the 2020s, 990 mm in the 2050s, and 1064 mm in the 2080s, while the lowest are in January as 70.3 mm

in the baseline (1961–1990), 74.4 mm in the 2020s, 76.5 mm in the 2050s, and 79.1 mm in the 2080s. Under the H3B2 scenario, monthly total CIR of five agricultural crops was highest in August as 905 mm in the baseline (1961–1990), 962 mm in the 2020s, 967 mm in the 2050, and 1061 mm in the 2080s, while the lowest water demands were in January as 68.6 mm in the baseline (1961–1990), 68.3 mm in the 2020s, 74.2 mm in the 2050s, and 79.4 mm in the 2080s. Figure 10 depicted the seasonal crop irrigation requirement of each individual crop in both selected scenarios. Under the H3A2 scenario, total CIR to produce sugarcane is likely the highest as 1879.8 mm year$^{-1}$ in the baseline (1961–1990), 1934.6 mm year$^{-1}$ in the 2020s, 2061.1 mm year$^{-1}$ in the 2050s, and 2219.3 mm year$^{-1}$ in the 2080s. While under the H3B2 scenario the highest rate of it might be 1842.6 mm year$^{-1}$ in the baseline (1961–1990), 1896.5 mm year$^{-1}$ in the 2020s, 1879.8 mm year$^{-1}$ in the 2050s, and 2162.4 mm year$^{-1}$ in the 2080s. The rank of water consumption rate of rice was 2nd as it consumed 1118.7 mm year$^{-1}$ in the baseline and in the future it may reach to 1143.5 mm year$^{-1}$ (2020), 1189.5 mm year$^{-1}$ (2050), and 1268.9 mm year$^{-1}$ (2080) under the H3A2 scenario. In case of the H3B2 scenario, water consumption rate of rice was 1111.5 mm year$^{-1}$ in the baseline (1961–1190) and in the future it may reach up to 1140 mm year$^{-1}$ (2020), 1177 mm year$^{-1}$ (2050), 1251.1 mm year$^{-1}$ (2080). All crops show significant rise in the water consumption rate due to rise in temperatures and decrease in rainfall. Under the H3A2 scenario, the average annual CIR of sugarcane, maize, cotton, rice, and wheat could be increased by 339 mm, 123 mm, 181 mm, 150 mm, and 45 mm, respectively in 2080 with respected to the baseline.

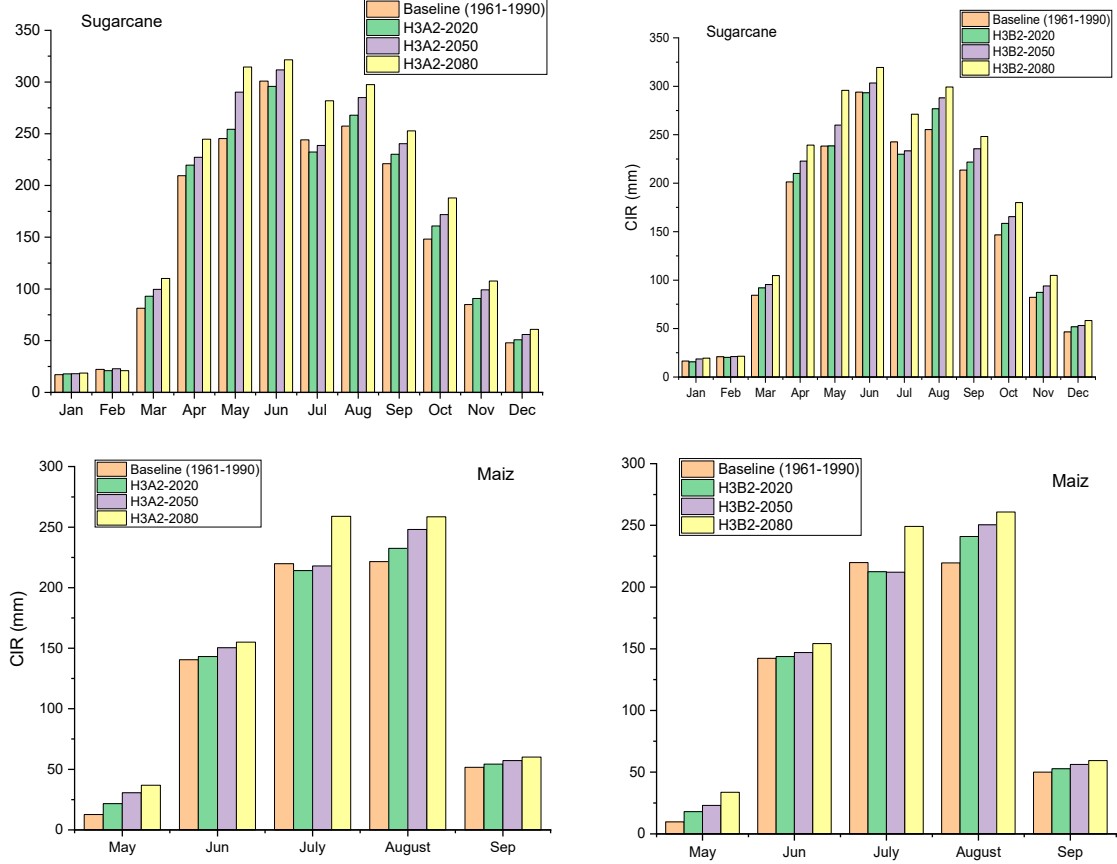

**Figure 8.** *Cont.*

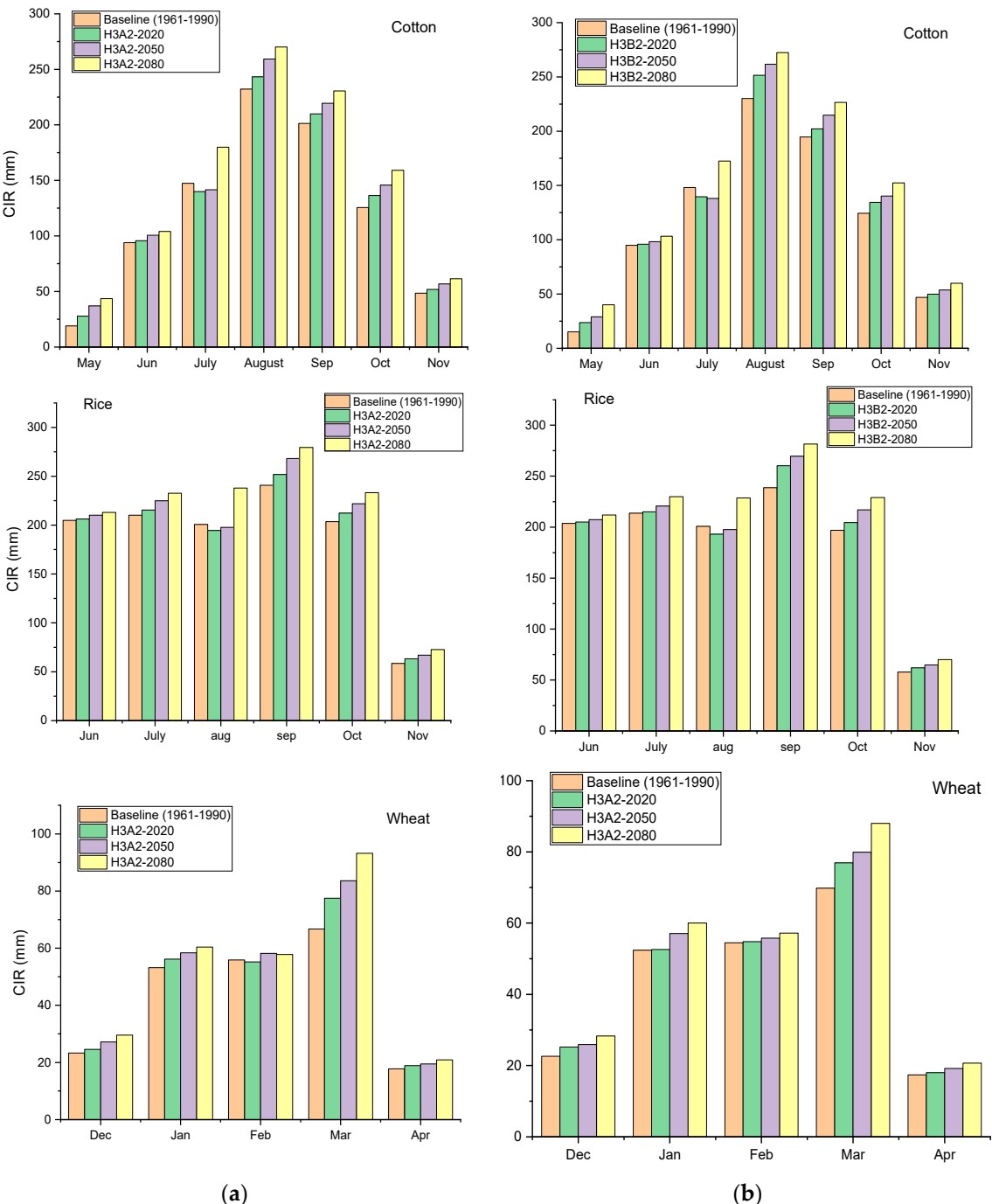

**Figure 8.** Monthly changes in crop irrigation requirement (CIR) of five agricultural crops, sugarcane, maize, cotton, rice, and wheat under (**a**) H3A2 and (**b**) H3B2 scenarios in Rechna Doab.

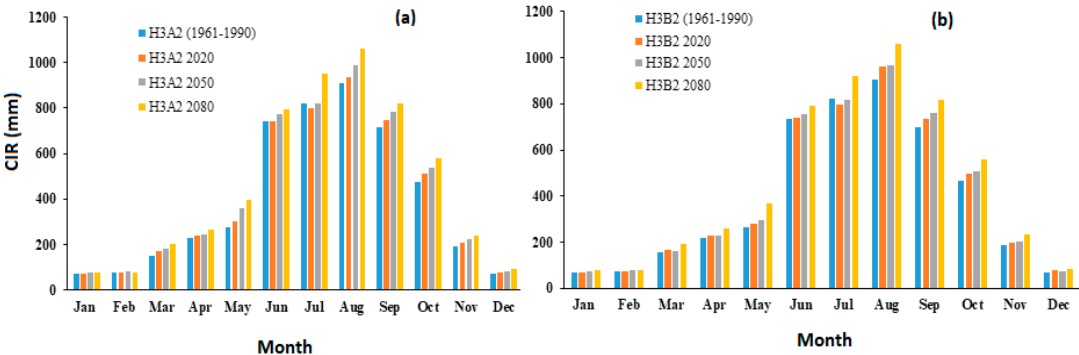

**Figure 9.** Monthly summation of CIR of five agricultural crops under (**a**) H3A2 and (**b**) H3B2 scenarios in Rechna Doab.

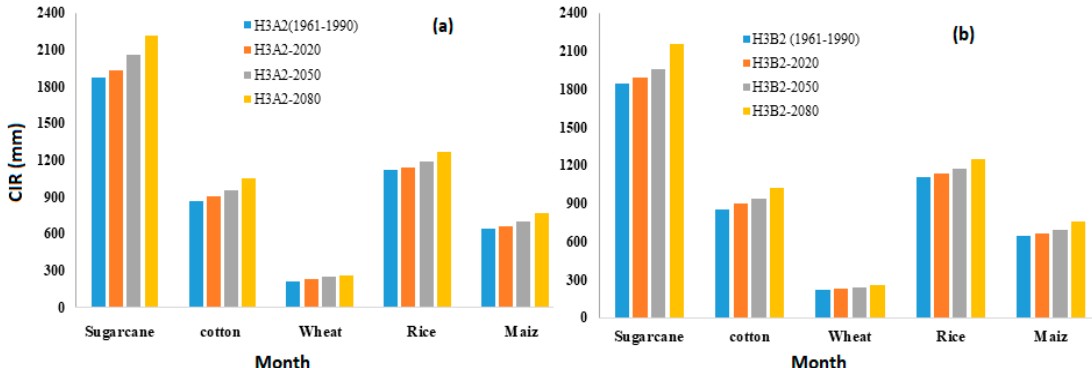

**Figure 10.** Changes in crop irrigation requirement (CIR) of different agricultural crops under (**a**) H3A2 and (**b**) H3B2 scenarios in Rechna Doab.

### 3.8. CIR under Changing Climate and Agriculture Cropping Patterns

In order to assess the impacts of both climate and agriculture planting area changes on CIR, future changes in total CIR were projected in three different sub-scenarios (1) S1: Changed climate with no change in agriculture land uses, (2) S2: Changed agriculture land use with no change in climate, (3) S3: Changed climate and agriculture land uses. Tables 6 and 7 indicate the changes in total CIR in H2A2 and under the H3B2 scenario under three sub-scenarios. Assuming the changing climate in the future (S1), it was observed that all crops may exhibit an increasing trend in CIR under all scenarios but with different magnitudes. Under the H3A2 scenario with sub-scenario (S1), CIR of sugarcane, cotton, wheat, rice, and maize may increase by 0.82, 0.28, 0.74, 1.27, and 0.13 BCM respectively in the 2080s relative to the baseline (1961–1990), and in the H3B2 scenario it may increase by 0.77, 0.26, 0.61, 1.18, and 0.13 BCM, respectively in the 2080s. Total CIR required to irrigate the five major crops may increase by 3.26 (A2) and 2.98 BCM (B2) in the 2080s with respect to the baseline scenario (1961–1990). Gradual rise in CIR may due to the rise in temperature and ET rate in the future. Assuming the changing planting area of crops in the future (S2), it was seen that sugarcane, wheat, and rice may consume more water due to the gradual rise in the planting area in the future while water application for cotton and maize may reduce because in the future their planting area may reduce. Under the H3A2 scenario with sub-scenario (S2), CIR of sugarcane, wheat, and rice may increase by 2.05, 1.73, and 9.59 BCM, respectively in the 2080s, and in the H3B2 scenario it may increase by 2.01, 1.73, and 9.53 BCM, respectively in the 2080s with respect to the baseline period. Total CIR required to irrigate the five major crops may increase by 12.12 BCM (A2) and 12.04 BCM (B2) in the 2080s with respect to the baseline scenario (1961–1990). Assuming with both changing climate and agriculture land use (S3), it was observed that the net volume of water to irrigate the crops may increase in future periods relative to the baseline. In the S3 scenario, total CIR required to irrigate the five major crops may

increase by 17.15 BCM (A2) and 16.62 BCM (B2) in the 2080s with respect to the baseline scenario (1961–1990). The S1, S2, and S3 scenarios clearly explain the individual impacts of both parameters (climate and agriculture land use) on net volumetric application of water for agriculture crops in the future. It can be seen that the S2 scenario may affect the CIR with greater magnitude compared to the S2 scenario. However, combined affects (S3) of both parameters have great implications for future CIR in Rechna Doab.

**Table 6.** Total CIR for different crops under the H3A2 scenario with three different sub-scenarios S1, S2, and S3. * Note: CSA indicated crop sown area.

| | | S1 | | | S2 | | | S3 | | |
|---|---|---|---|---|---|---|---|---|---|---|
| | | CIR | CSA | Total CIR = CIR*CSA | CIR | CSA | Total CIR = CIR*CSA | CIR | CSA | Total CIR = CIR*CSA |
| | Crops | mm year⁻¹ | 1000 ha | BCM | mm year⁻¹ | 1000 ha | BCM | mm year⁻¹ | 1000 ha | BCM |
| | | $mm\ year^{-1}$ | $1000\ ha$ | BCM | $mm\ year^{-1}$ | $1000\ ha$ | BCM | $mm\ year^{-1}$ | $1000\ ha$ | BCM |
| **2020s** | Sugarcane | 1934.6 | 241.55 | 4.6730263 | 1879.8 | 271 | 5.094258 | 1934.6 | 271 | 5.242766 |
| | cotton | 904.7 | 156 | 1.411332 | 867.7 | 120 | 1.04124 | 904.7 | 120 | 1.08564 |
| | Wheat | 232.4 | 1648.68 | 3.8315323 | 216.9 | 1900 | 4.1211 | 232.4 | 1900 | 4.4156 |
| | Rice | 1143.5 | 852 | 9.74262 | 1118.7 | 890 | 9.95643 | 1143.5 | 890 | 10.17715 |
| | Maize | 665.7 | 113 | 0.752241 | 646.4 | 87 | 0.562368 | 665.7 | 87 | 0.579159 |
| | Total | 4880.9 | 3011.23 | 20.410752 | 4729.5 | 3268 | 20.775396 | 4880.9 | 3268 | 21.500315 |
| **2050s** | Sugarcane | 2061.1 | 241.55 | 4.9785871 | 1879.8 | 308.5 | 5.799183 | 2061.1 | 308.5 | 6.3584935 |
| | cotton | 960.1 | 156 | 1.497756 | 867.7 | 180 | 1.56186 | 960.1 | 180 | 1.72818 |
| | Wheat | 246.9 | 1648.68 | 4.0705909 | 216.9 | 2250 | 4.88025 | 246.9 | 2250 | 5.55525 |
| | Rice | 1189.5 | 852 | 10.13454 | 1118.7 | 1400 | 15.6618 | 1189.5 | 1400 | 16.653 |
| | Maize | 704.5 | 113 | 0.796085 | 646.4 | 66 | 0.426624 | 704.5 | 66 | 0.46497 |
| | Total | 5162.1 | 3011.23 | 21.477559 | 4729.5 | 4204.5 | 28.329717 | 5162.1 | 4204.5 | 30.7598935 |
| **2080s** | Sugarcane | 2219.3 | 241.55 | 5.3607192 | 1879.8 | 351 | 6.598098 | 2219.3 | 351 | 7.789743 |
| | cotton | 1048.9 | 156 | 1.636284 | 867.7 | 60 | 0.52062 | 1048.9 | 60 | 0.62934 |
| | Wheat | 261.9 | 1648.68 | 4.3178929 | 216.9 | 2448 | 5.309712 | 261.9 | 2448 | 6.411312 |
| | Rice | 1268.9 | 852 | 10.811028 | 1118.7 | 1710 | 19.12977 | 1268.9 | 1710 | 21.69819 |
| | Maize | 769.4 | 113 | 0.869422 | 646.4 | 47 | 0.303808 | 769.4 | 47 | 0.361618 |
| | Total | 5568.4 | 3011.23 | 22.995346 | 4729.5 | 4616 | 31.862008 | 5568.4 | 4616 | 36.890203 |

**Table 7.** Total CIR for different crops under H3B2 scenario with three different sub-scenarios S1, S2, and S3. * Note: CSA indicated crop sown area.

| | | S1 | | | S2 | | | S3 | | |
|---|---|---|---|---|---|---|---|---|---|---|
| | | CIR | CSA | Total CIR = CIR*CSA | CIR | CSA | Total CIR = CIR*CSA | CIR | CSA | Total CIR = CIR*CSA |
| | Crops | $mm\ year^{-1}$ | $1000\ ha$ | BCM | $mm\ year^{-1}$ | $1000\ ha$ | BCM | $mm\ year^{-1}$ | $1000\ ha$ | BCM |
| **2020s** | Sugarcane | 1896.5 | 241.55 | 4.580996 | 1842.6 | 271 | 4.993446 | 1896.5 | 271 | 5.139515 |
| | cotton | 897.3 | 156 | 1.399788 | 854.1 | 120 | 1.02492 | 897.3 | 120 | 1.07676 |
| | Wheat | 227.5 | 1648.68 | 3.750747 | 216.7 | 1900 | 4.1173 | 227.5 | 1900 | 4.3225 |
| | Rice | 1140 | 852 | 9.7128 | 1111.5 | 890 | 9.89235 | 1140 | 890 | 10.146 |
| | Maize | 667.8 | 113 | 0.754614 | 641.5 | 87 | 0.558105 | 667.8 | 87 | 0.580986 |
| | Total | 4829.1 | 3011.23 | 20.19894 | 4666.4 | 3268 | 20.58612 | 4829.1 | 3268 | 21.26576 |
| **2050s** | Sugarcane | 1962.1 | 241.55 | 4.739453 | 1842.6 | 308.5 | 5.684421 | 1962.1 | 308.5 | 6.053079 |
| | cotton | 935.6 | 156 | 1.459536 | 854.1 | 180 | 1.53738 | 935.6 | 180 | 1.68408 |
| | Wheat | 237.9 | 1648.68 | 3.92221 | 216.7 | 2250 | 4.87575 | 237.9 | 2250 | 5.35275 |
| | Rice | 1177 | 852 | 10.02804 | 1111.5 | 1400 | 15.561 | 1177 | 1400 | 16.478 |
| | Maize | 688.8 | 113 | 0.778344 | 641.5 | 66 | 0.42339 | 688.8 | 66 | 0.454608 |
| | Total | 5001.4 | 3011.23 | 20.96287 | 4666.4 | 4204.5 | 28.08194 | 5001.4 | 4204.5 | 30.02252 |
| **2080s** | Sugarcane | 2162.4 | 241.55 | 5.223277 | 1842.6 | 351 | 6.467526 | 2162.4 | 351 | 7.590024 |
| | cotton | 1027 | 156 | 1.60212 | 854.1 | 60 | 0.51246 | 1027 | 60 | 0.6162 |
| | Wheat | 254.2 | 1648.68 | 4.190945 | 216.7 | 2448 | 5.304816 | 254.2 | 2448 | 6.222816 |
| | Rice | 1251.1 | 852 | 10.65937 | 1111.5 | 1710 | 19.00665 | 1251.1 | 1710 | 21.39381 |
| | Maize | 757.2 | 113 | 0.855636 | 641.5 | 47 | 0.301505 | 757.2 | 47 | 0.355884 |
| | Total | 5451.9 | 3011.23 | 22.53135 | 4666.4 | 4616 | 31.59296 | 5451.9 | 4616 | 36.17873 |

## 4. Discussion

Changing climate and agriculture expansion in the form of growing plantation area have greater implications for irrigated water supply [36]. This study analyzed the long-term changes in CIR (crop

irrigation requirement) caused by climate and agriculture land use changes. Past research has analyzed that minimum and maximum temperatures have increased in both seasons (winter and summer) throughout Pakistan, specifically central regions of Pakistan (Punjab) are becoming more warmer and dryer [78–80]. GCMs (general circulation models) can simulate future changes in climate. HadCM3 under two scenarios (A2 and B2) was used to capture climate change characteristics in the study region. Results of this paper indicate that under both scenarios (A2 and B2), temperatures (Tmax, Tmin) would increase in future periods (2020s, 2050s, and 2080s) while precipitation will decrease with respect the baseline scenario (1961–1990). These results indicate that climate in this region may likely be warmer and dryer in the future which may have greater implications for crop evapotranspiration and irrigation requirement. Reference ET is highly dependence on change in weather conditions, especially air temperatures. Various studies conducted in different regions have also indicated that reference ET has continuously increased due to gradual rise in temperatures [81–83]. Results of this study have shown that, in both climate change scenarios, monthly $ET_o$ is increasing gradually from January to the peak in July and then decreasing gradually up to December. The highest increase in reference ET is from April to September which is due to a hot and dry summer and low rainfall in these months as well as growth period of crops [84]. Long-term future changes in $ET_o$ have indicated that it would increase in future periods due to an increase in temperature, which would cause a gradual rise in irrigation requirement. This study predicted that the crop irrigation requirement of fiver major crops (rice, sugarcane, maize, wheat, and cotton) would increase in the future due to a gradual rise in ET in different months with different magnitudes. Monthly sum of CIR of five crops is found to be the maximum in August and minimum in January, which is related to ET and the growing stage. The reason why maximum water demand takes place in August is mainly due to the fact that three crops (sugarcane, cotton, and rice) out of five total crops have mid-season stage in August, resulting in more consumption of water. Sugarcane and rice consume more water as compared to other crops [85]. References [84,86] have analyzed impacts of climate change on crop water requirement of rice, sugarcane, maize, wheat, and cotton and their analysis also indicated that crop water demand will increase due to the rise in temperature in future scenarios.

Results of agriculture cropping patterns indicated that in future scenarios crop sown area of sugarcane, wheat, and rice may exhibit an increasing trend while cotton and maize may exhibit a reducing trend. This is because of the growing population demanding more food requirement to meet their needs. Past studies indicated that food and fiber requirement in Pakistan has largely increased as a result of rapid population growth, from 37.5 million in 1950 to 207 million in 2017 and is expected to reach 333 million in 2050 [26]. Wheat, sugarcane, maize, rice, and cotton are major food and fiber crops in Pakistan. In past years, the crops sown area of wheat, rice, and sugarcane has increased at a greater rate because these are staple food crops and extensively used in daily life, also the product of these crops are exported in other countries. Farmers are growing more crops on more land area to meet the demand of the population, therefore, land and water resources of Pakistan are tremendously under pressure to fulfill the growing needs of the population [27]. The study conducted by [87] indicated that the area under major crops (fooder, wheat, rice, sugarcane, cotton) has been increased in the past few years and will also increase more in the future to meet the food demand of a rapidly growing population in Pakistan. In Rechna Doab, the total irrigated area from 1.94 Mha in 1961 to 2.12 Mha in 1990 and cropping intensity has increased from 91% in 1961 to 131% in 1980 [28]. In order to analyze the net volume of water (Total CIR (m$^3$) = CIR×A) consumed by individual crops, the sown area of each crop was multiplied with the crop irrigation requirement of that crop. Assessment of individual impacts of both climate and agriculture cropping patterns on CIR is crucial, that is why future CIR is subjected to three further sub-scenarios S1 (changed climate), S2 (changed agriculture land uses), and S3 (changed climate and agriculture land uses). The climate change scenarios indicate that future CIR is likely to increase in the study area but at a less rate when compared with agricultural land use change scenario. However combined affects (S3) of both parameters have much severe implications for future CIR in Rechna Doab. This study serves as a first exploration on how the potential impacts

on environmental sustainability of agricultural expansion and intensification can be expressed over time, using a combination of approaches (land use and climate change scenarios). In order to gain a deeper insight into the impacts at a regional scale, future studies need to include spatial analysis and analyze additional climate change scenarios. Anyhow, this study has some limitations which may be the subject of future investigations. Firstly, Kc values of different agricultures crops were taken from FAO tables and relevant studies and these values were kept constant in the future periods to allow for a less complicated process and more comprehensive analysis. Secondly, agriculture land use changes in the form of crop sown area in future periods was just simulated using trend analysis based on the historical rate. In practice, agriculture expansion may fluctuate because of many factors ranging from population growth to socio-economic factors. Further attempts are required to simulate the crop sown area in the future based on socio-economic factors, foods demands, as well as their value addition in the market.

## 5. Conclusions

Irrigated agriculture is the largest water consuming sector in Rechna Doab. Growth of irrigated surfaces and the changing climate has placed new demands on the irrigated water supply. This study attempted to investigate current and projected changes in CIR under changing climate and agriculture expansion. This study predicted that in the future, the climate of Rechna Doab, Pakistan, will become warmer and dryer due to increasing temperatures and a decrease in rainfall and humidity. During calibration and validation of the SDSM model, high consistency of results with observed data indicated that the arid region of Rechna Doab is reliable for climate change prediction. The SDSM-HadCM3 A2 scenario had the greatest changes, namely an increase of 1.10 to 3.51 °C and 0.9 to 3.02 °C in the monthly means of Tmax and Tmin, respectively while in the B2 scenario it could be increased by 0.85–2.89 °C and 0.85–2.94 °C respectively in 2080 relative to the baseline (1961–1990). The gradual rise in temperature may have significant implications on crop water demand. When these increasing trends of temperatures were used to estimate future potential evapotranspiration using CROPWAT, increases in ET of the crops were found leading to elevated CIR. Among all agriculture crops, sugarcane was accounted as the major consumer of water, followed by rice which is at the 2nd rank. Similarly, projected changes in agriculture land use changes in the form of cropping patterns (crop sown area) depicted an increasing trend in the plantation area of sugarcane, wheat, and rice, while maize and cotton predicted a decreasing trend with respect to the baseline scenario. Overall, analysis presented that CIR is likely to increase under changing climate but at a less rate when predicted with the agricultural land use change scenario. However, combined impacts of both parameters (climate and agriculture land use changes) were found to be more severe for elevating CIR in the future period. In this study, simulated future irrigation requirement is an important asset to identify the emerging trends and support water management strategies.

**Author Contributions:** A.A. and W.Z. conceived this research. A.A. and Z.Z. performed the experiments, analyzed the results and wrote the paper under the guidance of W.Z. W.Z., Z.Z., and I.G. gave comments and modified the manuscript.

**Funding:** This research was financially supported by the National Key Research and Development Program of China (Project Nos. 2016YFA0602302 and 2016YFB0502502) and the National Natural Science Foundation of China (Project No. 41175088). Great thanks should be given to the Hadley Centre for making the Hadley Centre coupled model (HadCM3) available for public uses.

**Conflicts of Interest:** Authors declare no conflicts of interest.

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
