# Peer review of "Long-Term Perspective Changes in Crop Irrigation Requirement Caused by Climate and Agriculture Land Use Changes in Rechna Doab, Pakistan"

_water, doi:10.3390/w11081567_

Round 1

Reviewer 1 Report

The authors predicts the effect of climate change and land use change in a Pakistan region.

I suggest to revise the first part of the introduction section (until P2 L 13). This part is really slow to read . I suggest to shorten it. Moreover, Climate change is always related to anthropogenic activities. I suggest to remove it, a debate is ongoing about that, it is a judgment, and it is not related to the manuscript scope.

I believe that the weakest part of this manuscript is the land use change analysis. I suggest to remove it.

The authors assess land use change using only land use trends form the past. However crop plantation driver are policy and market. The farmer crops the crops he can sell. If people habit goes toward western habits, it is possible to attend the rise of wheat production and the reduction of rice. But it is hard to define what will happen in 50 years. The authors does not consider a change of Kc related to genetics improvement or  changes in technology (i.e. irrigation techniques) or the effect of higher temperature and the relative crop substitution. Moreover the cropped surface is set to stable as planted hectares.

I suggest acceptance with revisions.

Reviewer 2 Report

Reviewed publication depicts very interesting problems regarding the forecast of changes in water demand for crop irrigation using mathematical models.

Test methodology was designed appropriately and well explained. Test results were presented clearly and are very extensive.

In some fragments the text is overly extensive (repetition occurs). The authors could try to shorten less significant portions of text.

I suggest deleting following fragments:

1) Page 1, line 39 to page 2, line 2:

“Changing climate (due to anthropogenic activities) and agriculture (…) in future scenarios (2100) it will increase by 1.5 °C [10].”

2) Page 19, line 12 to 15:

“Changing climate (due to anthropogenic activities) and agriculture expansion (…) Indus Basin, Pakistan are accounted as world’s most vulnerable regions to climate change.”

Regarding this portion (page 19, line 15 to 22):

“Change in climate may have affects for number of sectors in Pakistan, (…) GCMs (General circulation models) can simulate future changes in climate.”

I suggest for it to be moved to introduction

Square brackets in the description of equation 3 should be changed to parentheses.

Author Response

Greatly thanks for your professional and constructive comments. It’s very nice of you to give us such excellent suggestions to help us improve the manuscript. All of the comments were carefully considered and replied here one by one.

Please have a look attached file all response have mentioned clearly point by point.

Sincerely appreciate your constructive and professional comments!

With our best regards!

Arfan Arshad and Wanchang Zhang
